# Functionalization of Silica with Triazine Hydrazide to Improve Corrosion Protection and Interfacial Adhesion Properties of Epoxy Coating and Steel Substrate

**Ayman M. Atta** [1,*], **Mona A. Ahmed** [2], **Ahmed M. Tawfek** [1] **and Ayman El-Faham** [1,3]

1   Department of Chemistry, College of Science, King Saud University, P.O. Box 2455, Riyadh 11451, Saudi Arabia; atawfik@ksu.edu.sa (A.M.T.); aelfaham@ksu.edu.sa (A.E.-F.)
2   Egyptian Petroleum Research Institute, Nasr City, Cairo 11727, Egypt; mona_chemist17@yahoo.com
3   Chemistry Department, Faculty of Science, Alexandria University, P.O. Box 426, Ibrahimia, Alexandria 12321, Egypt
*   Correspondence: aatta@ksu.edu.sa

**Abstract:** The chemical bonding of modified filler surfaces with coating networks is an advanced approach for improving the interfacial adhesion force of fillers with coating and substrate surfaces. In this respect, silica gel surfaces were activated and modified by grafting 1,3–dihydrazide-2,4,6-triazine onto hydroxyl groups of activated silica surfaces. The chemical structure, thermal stability and surface morphologies of the modified silica were investigated. The modified silica fillers were blended during the curing of the epoxy resin with the polyamine hardener. The data regarding the chemical structure and thermal stability of the cured epoxy networks in the presence of modified silica elucidated the chemical bonding of amine groups on the silica surfaces cured with the oxirane epoxy resin. Moreover, the incorporation of modified silica in surfaces with epoxy networks improved their adhesion with steel surfaces and enhanced the mechanical, thermal and anticorrosion characteristics of the epoxy to protect steel surfaces against seawater.

**Keywords:** steel coatings: silica; corrosion; epoxy coating; composites; triazine

## 1. Introduction

Among various organic coatings, epoxy coatings are widely used to protect steel substrates against corrosion due to their good barrier and mechanical properties [1–4]. Highly crosslinked epoxy networks produce microcracks and holes during the curing process on the steel surface. The presence of holes and cracks increases water vapor permeability and corrosive electrolytes, producing corrosive rusts as corrosion products, adhesion failure and cathodic delamination [5]. There are several techniques used to improve the adhesion of coatings on various metal substrates, such as removing contaminates on the steel surface (dust, oil, oxides/hydroxides and salt), treatment of the steel surface with chromate, phosphate washing and sol–gel-based silane coatings [6]. The toxicity of chromate and phosphate chemicals limits their application due to environmental concerns [6]. New approaches based on functionalization of environmentally friendly carbon derivatives, such as graphene, with new functional groups such as epoxide, carboxylic acids and amines have been recommended to promote the adhesion properties of steel surfaces with epoxy via chemical-covalent bonding between the steel substrate and epoxy coatings [7–9]. Moreover, chemical modification and functionalization of organic or inorganic fillers with epoxy or amine groups for bonding with epoxy networks and steel surface has also been recommended to improve performance of the epoxy coatings [10–13]. In this

respect, silicon-containing monomers and polymers, such as carbofunctional silane, polysiloxanes and polyhedral silsesquioxanes fillers have been used as eco-friendly multifunctional fillers to improve epoxy coating performance [14].

Silicone components have been used to increase weathering resistance, thermal, chemical, flame retardant and anticorrosive performance of epoxy coatings, even in harsh and aggressive conditions [15–20]. Incorporating siloxane, amine, hydroxyl and epoxy functional groups in the chemical structure of silicone-containing materials improves the curing of epoxy resin and enhances the adhesion, mechanical, thermal, durability and anticorrosive properties of the epoxy coatings with different substrate surfaces [14]. The presence of Si–O–Si significantly reduces water uptake of organic coatings with unreacted alkoxy groups on silica, reducing their water permeability, and improving their anticorrosive properties [15]. Moreover, the frangibility and thermal stability of epoxy coatings have been improved with the presence of organic silicones fillers [16–18]. The incorporation of organic compounds such as 1,3,5-triazine in the chemical structure of silicone-containing fillers has produced hyperbranched siloxane polymers [21,22]. They have been used to combine advantages of improved toughness, excellent dielectric properties, lower water absorption and high temperature resistance of epoxy coatings [23]. In our previous works [24,25], the presence of 1,3,5-triazine hydrazide as a capping agent for magnetite and silver nanoparticles improved their self-healing characteristics as fillers for epoxy coatings on steel surfaces. In this respect, the present work aims to activate silica gel surfaces by reacting them with 2, 4, 6-trichloro-1, 3, 5-triazine, followed by adding hydrazide groups. The goal of this modification is the chemical bonding of the activated silica gel with epoxy resin when cured with a polyamine hardener. The effect of modified silica with triazine hydrazide group contents on the curing, mechanical, thermal, adhesion and anticorrosive performances of the epoxy coatings applied on the steel substrate is investigated.

## 2. Materials and Methods

### 2.1. Materials

All chemicals were purchased from Aldrich–Sigma Co. and used as received. Silica gel (silica gel 60–230 mesh; SG) was activated (50 g) by refluxing with concentrated HCl (500 mL) for 24 h. The activated silica (AS) was filtered and washed by water then dried at 100 °C for 24 h. Cyanuric chloride (TCT; 98.0%) and hydrazine hydrate (80.0%) were used to modify the chemical structures of SG and AS. Tetrahydrofuran (THF), dichloromethane (DCM), ethyl acetate, butyl acetate, ethanol, toluene, acetonitrile and acetone were used as solvent. Commercial epoxy resin (SigmaGuard ™ CSF 650; 90 vol.%) and its polyamine hardener were purchased from SigmaKalon Group. Butyl acetate was added as thinner for epoxy resin. The recommended mixing weight ratio of 4:1 wt.% (epoxy resin: hardener) was used to obtain the epoxy coatings. Seawater was collected from Arabic gulf (Dammam, Kingdom of Saudi Arabia) with the total dissolved salt, $Na^+$, $Mg^{2+}$, $Ca^{2+}$, $Cl^-$ and $SO_4^{2-}$ concentrations of 45,000, 15,800, 1700, 500, 23,000 and 3200 ppm, respectively. Steel panels with the following chemical composition (in wt.%) of 0.14% C, 0.57% Mn, 0.21% P, 0.15% S, 0.37% Si, 0.06% V, 0.03% Ni, 0.03% Cr and Fe as balance were used as the substrate after they washed with xylene, soap and water to remove hydrophobic contaminates on the steel surfaces. Before coating application, the steel panels were blasted to produce rough surfaces (35 μm) using a blasting machine, washed with dry acetone and air-dried to remove any rust after blasting.

### 2.2. Preparation Methods

#### 2.2.1. Synthesis of Silica 2,4,6-triazinoxy-1,3-diaminohydrazide (STHc)

SG (30 g) and TCT (13.2 g, 72 mmol) were mixed together in DCM (200 mL) in a reaction flask equipped with a reflux condenser and heated for 2 h. The precipitate was separated by filtration, washed with DCM and dried in air to yield white solid (STC; yield: 33.8 g). The STC (30 g) was

suspended in acetonitrile (200 mL) and 30 mL of hydrazine hydrate was added. The reaction mixture was stirred under heating for 3–4 h, filtered, washed with acetonitrile and ethanol five times to ensure that the hazardous hydrazine and solvents were removed, and the powder was dried in air. The white slightly pink powder was precipitated to obtain STHc with yield of 33.4 g.

### 2.2.2. Synthesis of Activated Silica 2,4,6-triazinoxy-1,3-diaminohydrazide (STHa)

Activated silica gel (AS) was used instead of SG to react with TCT and hydrazine hydrate as illustrated in the previous section. In this respect, AS (16 g) and TCT (3.68 g, 20 mmol) was mixed together in dry dichloromethane (dry DCM, 200 mL) and stirred at room temperature for 2 h. The precipitate was filtered, washed with acetonitrile, ether and then dried to obtain white solid ASTC with yield 25.6 g, before drying in oven at 100 °C overnight. The activated silica ASTC (20 g) was suspended in acetonitrile (200 mL) and 40 mL of hydrazine hydrate was added. The reaction mixture was stirred under heating for 4 h, filtered washed with acetonitrile and ether, and dried in air to obtain STHa (loading 1.168 mmol TCT/g).

### 2.3. Characterization

The chemical structures of AS, SG, STHa and STHc were confirmed using Fourier transform infrared (FTIR; Perkin Elmer 1430) spectroscopy. The surface morphologies of AS, SG, STHa, STHc and their cured epoxy composites were investigated using a scanning electron microscope (SEM JEOL, JSM 5600 LV) (JEOL, Tokyo, Japan). Thermogravimetric analyses (TGA; Mettler Toledo, model: GA/DSC1, Basel, Switzerland) were carried under nitrogen flow rate of 60 mL.min$^{-1}$, in the temperature range 25–800 °C, with heating rate of 10 °C min$^{-1}$ to investigate the thermal stability of the AS, SG, STHa, STHc and their cured epoxy composites. The salt-spray resistance of the coated steel panels with epoxy and its blends with the AS, SG, STHa and STHc epoxy composites to seawater was investigated using a salt-spray cabinet (manufactured by CW Specialist Equipment Ltd., Leominster HR6 0LX, United Kingdom, model SF/450) at temperature of 35 °C.

### 2.4. Application of Coatings of the Steel Substrate

AS, SG, STHa and STHc were mixed with epoxy resin in the presence of butyl acetate as solvent using planetary centrifugal mixer (AR-100). The weight contents of AS, SG, STHa and STHc ranged from 1 to 5 wt.% related to the total weights of both epoxy resin and hardener. The epoxy composites were mixed with polyamine hardener according to the recommended mixing ration and sprayed on blasted and clean steel panels (roughness 35 μm) with an average dry-film thickness of 100 μm. The curing of the sprayed films was carried out at 120 °C for 4 h.

### 2.5. Mechanical Properties of Modified Epoxy Coatings

An hydraulic pull-off adhesion tester in the range of 0–25 MPa was used to determine the adhesion strengths of the cured epoxy films with the steel substrate according to ASTM D-4541. The abrasion resistance was determined using ASTM D-4060 (in terms of weight loss) and applied CS-17 wheels for 5000 cycles. The hardness of coated films was evaluated according to ASTM D-3363, using Erichsen hardness test pencil (model 318S with scratching force in the range of 0.5–20 N. The flexibility of the epoxy coatings was evaluated from bending and impact tests using conical mandrel and impact tester, according to ASTM D-522 and ASTM D-2794, respectively.

### 2.6. Corrosion Resistance of Modified Epoxy Coatings

The salt-spray, or corrosion resistances, of epoxy coatings were estimated according to the ASTM B117 procedure using seawater. Seawater was used instead of NaCl (5 wt.%) to study the effect of aggressive different salts cations, such as $Na^+$, $Mg^{2+}$ and $Ca^{2+}$—or their soluble anion—on the epoxy film durability in the marine environment. The salt fog and the humidity inside cabinet was maintained

at 98% at temperature of 35 ± 2 °C. The corrosion under epoxy coatings was evaluated according to ASTM D-1654, based on blistering, rusting and loss of adhesion.

## 3. Results and Discussion

The present work aims to modify the chemical structure of SG and AS by reacting with TCC, followed by reacting the remained chlorides of TCC with hydrazine, as reported in the experimental section and Scheme 1. It has been previously reported that SG can be activated with HCl (4 M) after drying at 150 °C for 3 h to remove the physically bonded water to remove the metal traces and to add more silanol groups (Si-OH) on the SG surface [26]. The silanol groups may be functionalized with amino groups by reacting STC with the hydrazine (Scheme 1). The SG and AS were reacted with TCC and hydrazine to obtain STHc and STHa, respectively.

**Scheme 1.** Synthesis of STHc and STHa.

### 3.1. Characterization of STHc and STHa

The chemical structures of AS, STHc and STHa were elucidated from FTIR spectra represented in Figure 1a–c. The activation of SG with HCl and drying produced more hydroxyl groups, as confirmed by the peak appearing at 3200–3700 cm$^{-1}$ (–OH stretching vibration); the disappearance of the shoulder peak at 3750 cm$^{-1}$ suggests that the hydroxyl group of water combined physically on the AS surface (Figure 1a). The presence of Si–O–Si and Si–OH groups in all spectra (Figure 1a–c) was confirmed from the appearance of bands observed around 1072 and 964 cm$^{-1}$ that referred to their stretching and bending vibrations, respectively. The bands at 465 and 815 cm$^{-1}$ were also attributed to Si–O bending vibration and appeared in all spectra (Figure 1a–c). The concentration of Si-OH groups was higher in STHc than STHa, as confirmed from the lower intensity bands at 964 cm$^{-1}$ (Si–OH) of STHa (Figure 1c) than STHa (Figure 1b) in the FTIR spectra, respectively. This suggests that the reaction of TCC and hydrazine with Si–OH groups was increased with the AS surface [27]. The appearance of two new bands at 1650 and 1567 cm$^{-1}$ were also attributed to C = N- and NH$_2$ stretching and the bending vibration of triazine hydrazide groups (Scheme 1) in the spectra of STHc (Figure 1b) and STHa (Figure 1c), respectively.

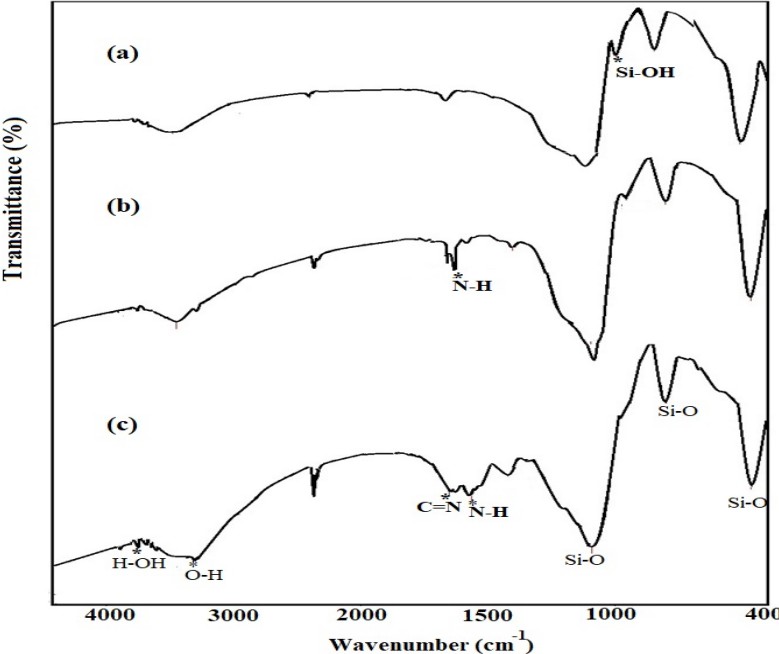

**Figure 1.** Fourier transform infrared spectroscopy (FTIR) spectra of (**a**) AS, (**b**) STHc and (**c**) STHa.

The surface morphologies of SG, AS, STHc and STHa were investigated from SEM photos in Figure 2a–d. The SG shows uniform, delicate, relatively smooth and dense morphology (Figure 2a) that changed to rough and much larger particle sizes with activation AS (Figure 2b). Moreover, more uniform with few pores and rough morphology are observed in STHc (Figure 2c) more than STHa (Figure 2d), respectively. This suggests that the pores were repaired after modification of AS with triazine hydrazide group, due to formation of intramolecular hydrogen bonding of triazine hydrazide groups [28].

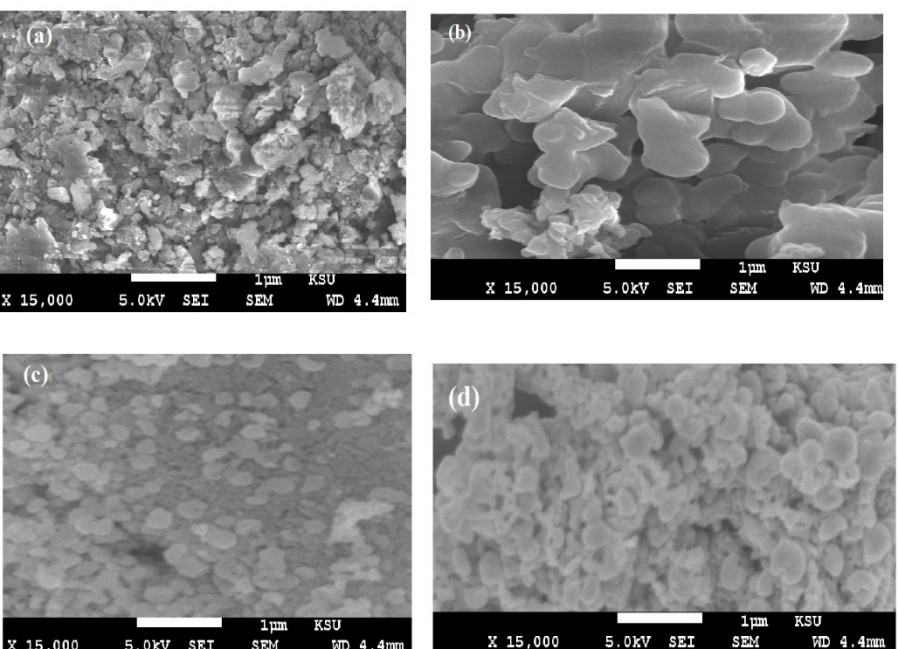

**Figure 2.** Scanning electron microscope (SEM) images of (**a**) SG, (**b**) ASG, (**c**) STHc and (**d**) STHa.

The thermal stabilities of SG, AS, STHc and STHa were evaluated from TGA thermograms represented in Figure 3. Only AS had a small weight loss around 100 °C (3–5 wt.%), corresponding to

the release of physically bonded water. The STHc thermograms (Figure 3) show higher thermal stability than STHA and lower 1,3,5-triazine hydrazide content, as confirmed from the initial degradation temperature (IDT) at 300 °C and the remained residual (RS; 70 wt.%) at 800 °C, which was attributed to non-degraded silica. The IDT and RS of STHa values were 230 °C and 53 wt.%, respectively (Figure 3). The difference between the RS values of STHa and STHc suggests that the triazine hydrazide 47 and 30 wt.% reacted on the STHa and STHc surfaces, respectively. The lower thermal stability of STHa than STHc may be attributed to the degradation and loss of $NH_3$, due to the higher content of hydrazide group of STHa than STHc [29].

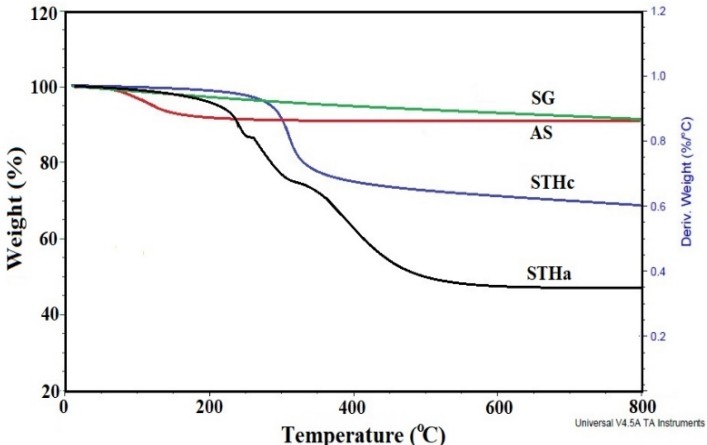

**Figure 3.** Thermogravimetric analyses (TGA) thermograms of SG, ASG, STHc and STHa.

### 3.2. Curing of STHc and STHa with Epoxy/Polyamine System

Different weight ratios of SG, AS, STHc and STHa ranged from 1 to 5 wt.% were blended with epoxy resins (based on bisphenol-A diglycidyl ether; DGEP) and cured with polyamine hardener as reported in the experimental section. The proposed curing mechanism is represented in Scheme 2. It was expected that the amine groups of the modified silica and 2,4,6-triazine dihydrazide (STHa and STHc) would react with the epoxy groups during the curing process to produce more crosslinked networks having higher hydroxyl group content [30]. Accordingly, the presence of more triazine hydrazide groups in the chemical structures of STHa than STHc increased their linking of STHa with the epoxy matrix rather than that modified with SG and AS. The dispersion of SG, AS, STHc and STHa in the butyl acetate was evaluated by DLS as represented in Figure 4a–d. Their particle size diameters and polydispersity index (PDI) were measured and listed in the Figure 4a–d. It was found that the activation of SG with HCl increased AS particle sizes and reduced their dispersion in butyl acetate solvent via increasing PDI from 0.635 (SG; Figure 4a) to 1.435 (AS; Figure 4b). These data agree with the SEM images represented in Figure 2. The increasing diameter of SG from 13 μm (Figure 4a) to 29.5 μm (Figure 4b) may be attributed to the rough morphology of AS (Figure 2) that increased their swelling with the butyl acetate solvent. It was also noticed that the modification of SG and AS with triazine hydrazide STHc and STHa improved their dispersion in the butyl acetate solvent via lowering PDI from 0.635 to 0.291 (Figure 4c) and from 1.435 to 0.765 (Figure 4d), respectively. The STHc showed more uniform and monodispersed particles than STHa that may be attributed to the formation of intramolecular hydrogen bonds occurred among the amine groups of triazine hydrazide and their intermolecular hydrogen bonding with butyl acetate solvent [28]. These interactions were retarded by increasing 2,4,6-triazine hydrazide on the surfaces of STHa, which increased their polydispersity due to more restriction in the free rotation of hydrazide groups [31].

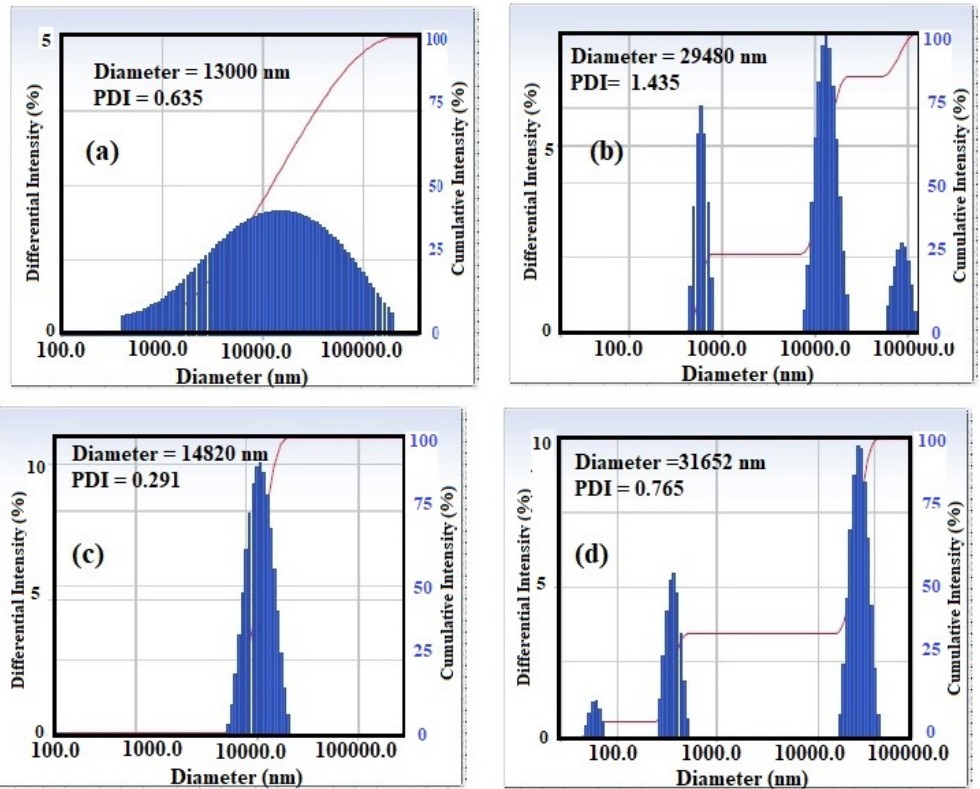

**Scheme 2.** Curing of STHa with epoxy/polyamine hardener.

**Figure 4.** Dynamic light scattering of dispersed (**a**) SG, (**b**) ASG, (**c**) STHc and (**d**) STHa in polyamine hardener solution.

The chemical linking of STHc and STHa with the epoxy networks investigated from FTIR spectra of their composites with epoxy/polyamine hardener as represented in Figure 5a–c. The appearance of new bands at 1675 and 1580 cm$^{-1}$ in the spectra of epoxy composites with 5 wt.% of STHc (Figure 5a) and STHa (Figure 5b) may be attributed to C = N- and NH$_2$ stretching and bending vibration of triazine hydrazide groups, indicating their incorporation into epoxy composites. Increasing the O-H groups intensity (appeared at 3450 cm$^{-1}$) in the presence of STHa (Figure 5b) over that obtained in the presence of STHc (Figure 5b), and absence of STHc or STHa (Figure 5c) elucidates the curing of amine groups of triazine hydrazide with oxirane, as represented in Scheme 2.

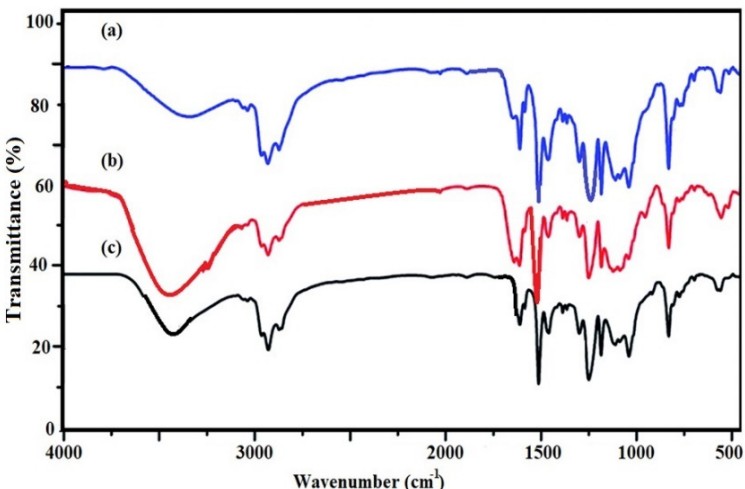

**Figure 5.** FTIR spectra of the cured epoxy/polyamine hardener in the presence (**a**) STHc (5 wt.%), (**b**) STHa (5 wt.%) and (**c**) without filler.

The effect of the wt.% of SG, AS, STHc and STHa on the surface morphology of the cured epoxy/polyamine hardener system was evaluated from SEM images (Figure 6a–e). The curing of epoxy with polyamine hardener produced epoxy with cracked and layered surface morphologies (Figure 4a) in the absence of fillers (blank), due to the formation of higher-crosslinking density epoxy/polyamine networks [32,33]. The activation of SG to AS changed the surface morphologies of cured epoxy in the presence of SG (Figure 6b) from skin to rough porous surfaces when blended with AS (Figure 6c). Increasing SG and AS up to 5 wt.% led to form filler aggregates on the epoxy surfaces (Figure 6b,c). Modification of SG and AS with triazine hydrazide produced better unique compatible surface morphologies without aggregation (Figure 6d,e) as result of stronger physical interactions through hydrogen bonds between amino groups and siloxane linkages [34]. The appearance of pores and holes in the SEM images of the cured epoxy in the presence of STHa (Figure 6d) may be attributed to the formation of high crosslinking densities of the cured epoxy matrix, due to higher triazine hydrazide contents on the surface of AS more than SG [35].

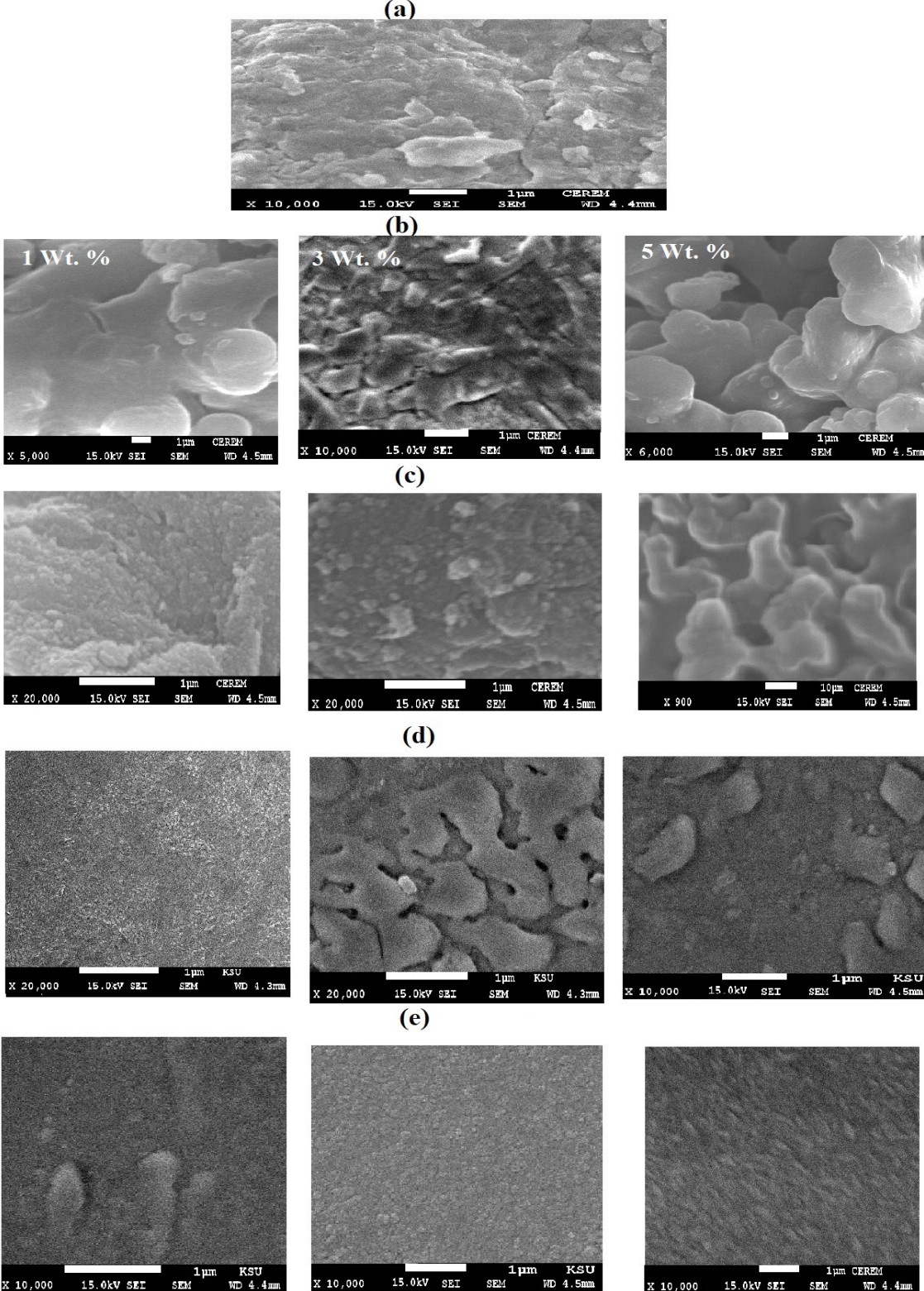

**Figure 6.** SEM images of the cured epoxy/polyamine hardener different wt.% of filler. From left to right 1, 3, 5 wt.% of (**a**) absence of filler, (**b**) SG, (**c**) AS, (**d**) STHa and (**e**) STHc.

The thermal characteristics of epoxy coatings blended with AS, STHa and STHc were investigated from TGA-DTG thermograms represented in Figure 7a–d. It was noticed that the degradation of epoxy networks in absence of silica hybrid (Figure 7a) showed two degradation steps. The first started from

180 to 350 °C to lose 60 wt.% from its initial weight. The second degradation step ended at 620 °C with residual weight 3 wt.%. The epoxy blends with 5 wt.% of AS thermograms (Figure 7b) showed also two degradation steps, but the IDT of epoxy networks increased from 180 to 250 °C and the RS at 750 °C increased up to 8 wt.%. The IDT of the epoxy network in the presence of STHa (Figure 7c) and STHc (Figure 7d) increased to 400 and 387 °C, respectively. Moreover, the RS values for the epoxy network cured in the presence of STHa (Figure 7c) and STHc (Figure 7d) increased to 22 and 15 wt.%, respectively. Increasing of IDT and RS values of the epoxy networks in the presence of STHa and STHc confirmed their chemical linking with epoxy matrix during the curing, and suggested successful incorporation of STHa and STHc into hybrid coatings [36–38].

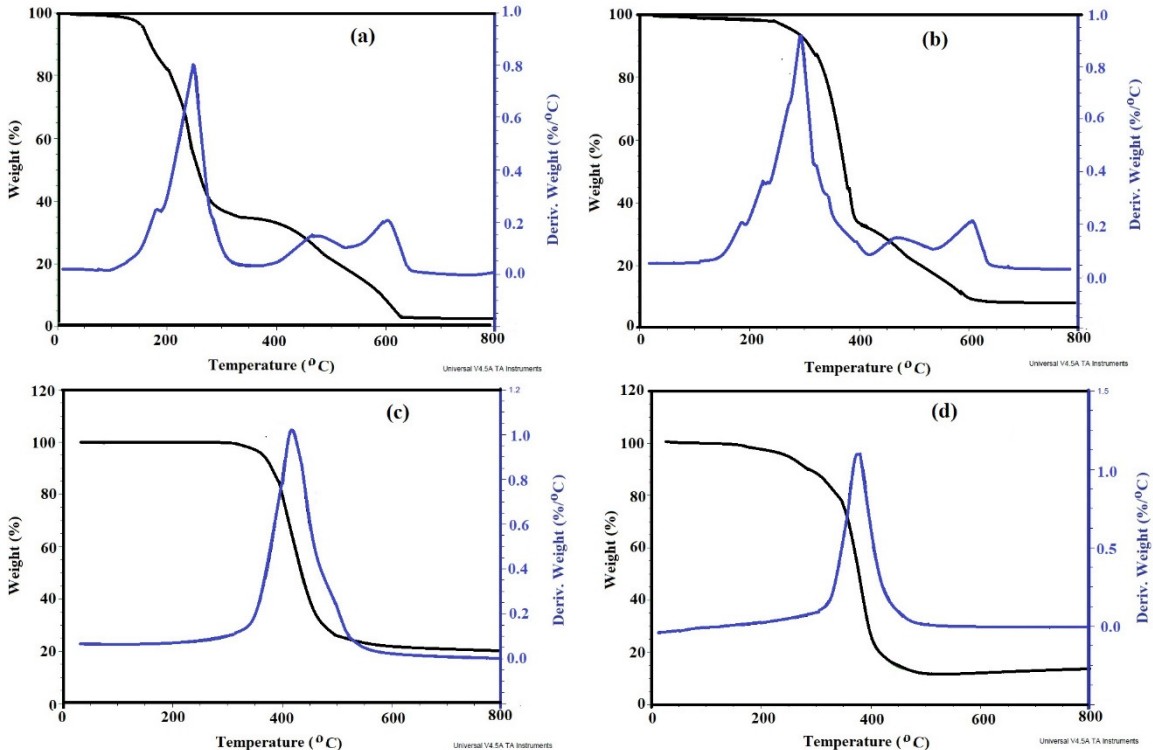

**Figure 7.** TGA-DTG thermograms of the cured epoxy/polyamine hardener in the presence 5 wt.% of (**a**) SG, (**b**) AS, (**c**) STHc and (**d**) STHa.

### 3.3. Adhesion and Mechanical Properties of Epoxy

One of the most drawbacks of using fillers in polymer composites is the lowering of interfacial adhesion of the fillers between organic polymers and the substrate surfaces [39,40]. The present study found that silica hydroxyl groups that were modified with cyclic triazine to improved interaction of silica with both epoxy networks and steel surfaces. Moreover, the presence of triazine and hydrazide groups on the silica surfaces assisted its chemical linking with epoxy networks during the curing with polyamine hardener (Scheme 2) to produce more hydroxyl groups that increased their interactions with the steel surfaces. It is also predicted that the presence of triazine hydrazide on the silica filler will promote their adhesion with the steel surface by chelation of amine lone pairs and Fe with coordination bonds [41]. It has been previously reported that modification of silica with hyperbranched poly (ethyleneimine) may increase their adhesion characteristics of the epoxy resin through the physically interaction of hydroxyl groups of silica that chemically react with the epoxide groups [42]. The interfacial adhesion bonds of SG, AS, STHc and STHa with epoxy networks and steel surfaces was examined from pull-off hydrolytic tester. The mechanical properties of coatings (impact resistance, pencil hardness, bending and abrasion resistance) were also are measured (Table 1). The adhesion data confirmed that STHc and STHa have a higher adhesion strength with the steel surfaces than AS

at different wt.% (Table 1). The epoxy networks blended with AS have a higher adhesion strength than SG and blank epoxy coatings. The enhancement of steel surface adhesion with the epoxy STHa composites are attributed to the higher triazine hydrazide contents as confirmed from FTIR spectra (Figure 1) and thermal stability curves (Figure 3). The high triazine hydrazide content on the silica surfaces of STHa generated more hydroxyl groups during curing with oxirane (Scheme 2). In addition, the bonding of triazine hydrazide on the steel surface through hydrogen bonding are responsible for high adhesion strength. The presence of –Si–OH and –Si–NH–$NH_2$ improved their bonding with the –OH groups presented on the steel surface [5]. Mechanical properties such as impact and hardness strength over SG and blank epoxy were also improved and increasing presence of STHc, STHa and AS. The presence of rough surface morphologies increased the impact resistance of AS more than SG, which was affected by increasing their wt.% due to aggregation (Figure 5). The presence of –NH–$NH_2$ groups of STHa and STHc changed the adhesion mechanism of epoxy coating on the steel surface from a mechanical interlocking state to a chemical mechanical form, via creating covalent bonds with –NH. These interactions increased the diffusion of STHa and STHc into the epoxy film, making strong mechanical interactions [5]. The lowering of abrasion resistance of epoxy coatings with STHa more than STHc may be attributed to the presence of cracks, as elucidated from SEM images [43].

**Table 1.** Mechanical tests of the cured epoxy resins modified with different weight percentages of SG, AS, STHa and STHc.

| Sample | Weight Contents (Wt.%) | Impact Test (Joule) | Hardness (Newton) | Pull of Test (MP) | Abrasion Resistance mg/1kg Weight for 5000 Cycles Weight Loss (mg) |
|---|---|---|---|---|---|
| blank | 0 | 5.0 ± 0.2 | 8.0 ± 0.8 | 5.0 ± 1.3 | 85 ± 2 |
| Epoxy/SG | 1.0 | 6.0 ± 0.5 | 10.4 ± 0.2 | 5.5 ± 0.5 | 40 ± 1 |
| | 3.0 | 6.3 ± 0.1 | 11.3 ± 0.3 | 6.4 ± 0.2 | 45 ± 2 |
| | 5.0 | 6.0 ± 0.2 | 9.5 ± 0.1 | 5.4 ± 0.2 | 49 ± 2 |
| Epoxy/AS | 1.0 | 10.0 ± 0.8 | 12.3 ± 0.1 | 7.4 ± 0.5 | 10 ± 1 |
| | 3.0 | 14.0 ± 0.8 | 14.6 ± 0.4 | 8.5 ± 0.4 | 8 ± 1 |
| | 5.0 | 12.0 ± 1.2 | 13.8 ± 0.3 | 8.4 ± 0.7 | 4 ± 0.2 |
| Epoxy/STHa | 1.0 | 12.0 ± 0.7 | 14.3 ± 0.5 | 15.5 ± 1.1 | 15 ± 2 |
| | 3.0 | 18.0 ± 0.3 | 17.2 ± 0.1 | 22.8 ± 1.9 | 4 ± 1 |
| | 5.0 | 16.0 ± 0.9 | 15.3 ± 0.5 | 20.3 ± 1.1 | 15 ± 3 |
| Epoxy/STHc | 1.0 | 14.0 ± 1.4 | 15.3 ± 0.4 | 12.3 ± 1.5 | 20 ± 2 |
| | 3.0 | 20.0 ± 1.5 | 18.4 ± 0.5 | 16.4 ± 0.7 | 10 ± 1 |
| | 5.0 | 21.0 ± 1.1 | 20.2 ± 0.2 | 18.5 ± 0.8 | 8 ± 1 |

A salt-spray resistance test has been recommended as an accelerated corrosion test to evaluate the diffusion of salt and water into the protective coating surfaces to steel surfaces [44]. It is well-known that the aggregation of fillers and lower interaction forces existing between the filler and coating matrix will reduce the barrier properties a mechanical properties of coatings. The present work aims to solve these drawbacks via formation of strong chemical covalent bonding between filler surfaces and the epoxy networks via the presence of coordination sites to chelate the epoxy coatings with the steel surface (Scheme 2). The chelation of coatings with a steel substrate forms a relatively thin uniform film which can act as barrier layer against corrosive seawater ions, hindering corrosion reactions on steel surfaces. Consequently, the epoxy coated steel panels were subjected to salt-spray tests for different exposure periods and evaluated according ASTM B17. The data from the salt-spray tests of the coated panels with the epoxy composites in the absence and presence of SG, AS, STHa and STHc are summarized in Table 2. Their photos were represented in Figure 8a–e. The data from epoxy blank and epoxy composites with SG (Table 2 and Figure 8a,b) showed the formation of blister and the corrosion initiated on the scratched surfaces after exposure to 500 and 650 h, respectively. The

defect area increased with lowering and increasing the SG wt.% to more than 3 wt.%. The same result occurred when silica was used as filler for the epoxy/polyamine hardener system [45]. The epoxy composites with AS photos (Figure 8c) showed no propagation of corrosion and blister formation with using 3 wt.% of AS after exposing of 750 h. However, the case of AS-embedded epoxy composites with 1 and 5 wt.% confirmed the initiation of corrosion on the scratched area on completion of 720 h exposure in seawater salt-spray. The corrosion initiation and propagation of the steel surfaces in seawater consisted of anodic dissolution of Fe and cathodic reduction of $O_2$ and $H_2O$ that was retarded due to the barrier of the residual hydroxyl groups linked with the epoxy organic coatings and steel substrate [46]. The removals of the residual linking of the hydroxyl and organic groups by environment produced some defects in the microstructure of the epoxy thin film and propagate crack formation. The rough structure of AS (Figure 6c) increased its corrosion rate when subjected to the corrosive environment [47]. The presence of STHa during the curing of epoxy produced some defect such as nanoholes and cracks in their surface morphologies (Figure 6d) due to the increasing of triazine dihydrazide contents. The presence of nanocracks increased the diffusion of salt and water through epoxy composites to initiate the corrosion of steel. The anticorrosion performance of epoxy/STHa was increased up to exposure time of salt-spray more than 1200 h (Figure 8d and Table 2). The STHc/epoxy composites showed excellent corrosion resistance after exposing more than 1750 hours (Figure 8e and Table 2). The corrosion protection of all epoxy composites was lowered with the increasing of the AS, STHc and STHa more than 3 wt.%; due to agglomeration within the epoxy coatings. The aggregation of fillers enhances the diffusion of corrosive salt. Accordingly, the chemical reaction of amine groups of triazine dihydrazide with epoxy/polyamine matrix and their chelation with the steel surfaces led to higher barrier properties of epoxy coatings in the case of STHa and STHc [48]. Moreover, both STHa and STHc epoxy composites had stronger adhesion strengths (Table 1) and provided high corrosion protection performance [49–52]. The lower dispersion of STHa in epoxy/polyamine hardener than that of STHc led to formation of cracks and defects that decreased the corrosion resistance of epoxy composites, due to agglomeration and hydrogen bonding of STHa with more triazine dihydrazide groups [53].

**Table 2.** Salt-spray resistance of the epoxy resins modified with different weight percentages of epoxy composites in the absence and presence of different weight percentages of SG, AS, STHa and STHc.

| Epoxy Composites | Weight Contents (wt.%) | Exposure Time (hours) | Disbonded Area % | Rating Number (ASTM D1654) |
|---|---|---|---|---|
| blank | 0 | 500 | 20 | 5 |
| Epoxy/SG | 1.0 | 650 | 5 | 7 |
| | 3.0 | 650 | 3 | 8 |
| | 5.0 | 650 | 5 | 7 |
| Epoxy/AS | 1.0 | 750 | 5 | 7 |
| | 3.0 | 750 | 3 | 8 |
| | 5.0 | 750 | 4 | 8 |
| Epoxy/STHa | 1.0 | 1750 | 6 | 7 |
| | 3.0 | 1750 | 3 | 8 |
| | 5.0 | 1750 | 4 | 8 |
| Epoxy/STHc | 1.0 | 2000 | 5 | 7 |
| | 3.0 | 2000 | 1 | 9 |
| | 5.0 | 2000 | 3 | 8 |

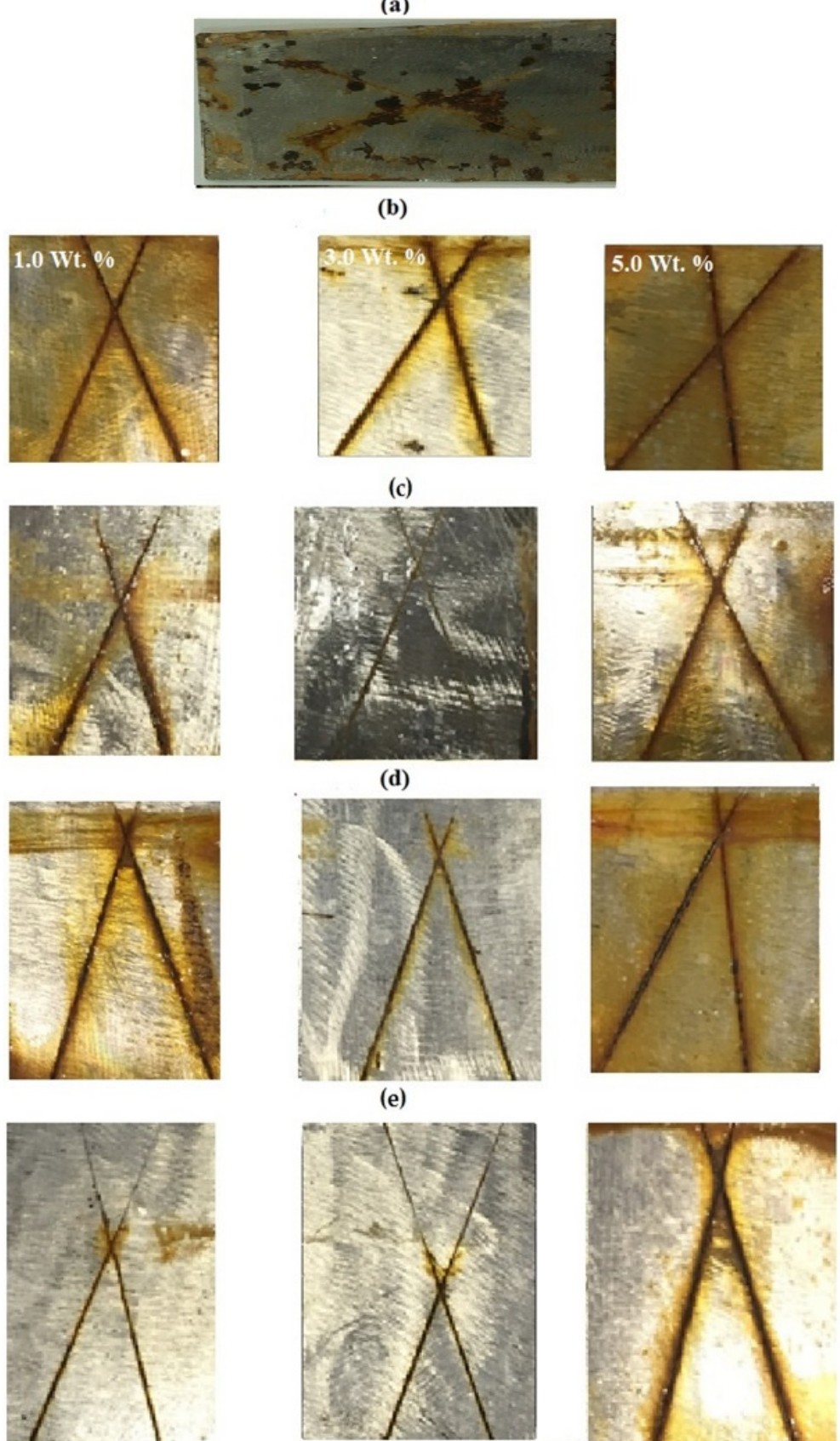

**Figure 8.** Salt-spray resistance of the cured epoxy coatings at different exposure times (**a**) epoxy blank after 500 h, (**b**) epoxy modified with SG after 650 h, (**c**) epoxy modified with AS after 750 h, (**d**) epoxy modified with STHa after 1750 h and (**e**) epoxy modified with STHa after 2000 h at 35 °C.

## 4. Conclusions

New modifications for activated silica gel with incorporation of dihydrazide 2,4,6-triazine were carried out to improve the adhesion, mechanical, thermal and anticorrosion performance of epoxy/silica composites on steel surfaces. The chemical structure, thermal characteristics and surface morphologies of silica grafts based on 2,4,6-triazine dihydrazide—designated as STHc and STHa—were investigated. The data show that activation of silica with dihydrazide 2,4,6-triazine changed the morphology to rough and porous particles. Thermal stability data indicated that STHa and STHc were modified with triazine hydrazide (wt.%), 47 and 30 wt.%, respectively. The particle sizes of STHc showed more uniform and monodispersed particles than STHa, a trait that can be attributed to the formation of intramolecular hydrogen bonds occurring among the amine groups of triazine hydrazide—and their intermolecular hydrogen bonding with amine groups of polyamine hardener. These interactions were retarded with increasing triazine hydrazide content on the STHa surfaces. The thermal stability data of epoxy composites with STHa and STHc confirmed their chemical linking with epoxy matrix during the curing and suggested successful incorporation of STHa and STHc into hybrid coatings. Moreover, both STHa and STHc epoxy composites had stronger adhesion strengths and provided high corrosion protection performance. The lower dispersion of STHa in epoxy/polyamine hardener than STHc led to formation of cracks and defects, so the corrosion resistance of epoxy composite coating was decreased.

**Author Contributions:** Conceptualization, A.M.A, A.E.-F., M.A.A. and A.M.T.; methodology, M.A.A. and A.M.T.; software, A.M.T.; validation, A.M.A, A.E.-F., M.A.A. and A.M.T.; formal analysis, A.M.A, A.E.-F., M.A.A. and A.M.T. investigation A.M.A, and A.E.-F.; resources, A.M.A, A.E.-F., M.A.A. and A.M.T.; data curation, A.M.A, A.E.-F., M.A.A. and A.M.T.; writing—original draft preparation, A.M.A.; writing—review and editing, A.M.A.; visualization, A.M.A, A.E.-F., M.A.A. and A.M.T.; supervision, A.M.A, and A.E.-F.; project administration, A.M.A.; funding acquisition, A.M.A. All authors have read and agreed to the published version of the manuscript.

**Funding:** This research was funded by King Saud University, researchers supporting project number (RSP-2019/63), King Saud University, Riyadh, Saudi Arabia.

**Acknowledgments:** The authors acknowledge King Saud University, researchers supporting project number (RSP-2019/63), King Saud University, Riyadh, Saudi Arabia for funding support.

**Conflicts of Interest:** The authors declare no conflict of interest.

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
