# Peer review of "Functionalization of Silica with Triazine Hydrazide to Improve Corrosion Protection and Interfacial Adhesion Properties of Epoxy Coating and Steel Substrate"

_coatings, doi:10.3390/coatings10040351_

Round 1
Reviewer 1 Report
ear authors
My comments are added (highlighted) in the file given as an attachment.
The main concern are:
- the procedure contains a lot of hazardous solvents (it is not in agreement to find an environmentally sustainable replacement to hazardous chromates),
- low quality of the figures
- salt spray testing (probably the test was performed with 5 % NaCl and not with seawater)
- the mechanism of network formation (bonding of epoxy groups) is only predicted (it was not confirmed by FTIR or other spectroscopic technique, SEM/EDS).
However, the manuscript can be accepted after these minor comments/revisions.
Regards

Author Response
- The procedure contains a lot of hazardous solvents (it is not in agreement to find an environmentally sustainable replacement to hazardous chromates)
- Answer: The low solubility of triazine and dispersion of silica ffected by selecting the greener solvent.
- low quality of the figures
- Answer: he quality of figures improved
- salt spray testing (probably the test was performed with 5 % NaCl and not with seawater)
- Answer: the using of sea water to test the coating durability was selected as aggressive marine environments that that aqueous solution in the presenceof NaCl 5 Wt. %.
- the mechanism of network formation (bonding of epoxy groups) is only predicted (it was not confirmed by FTIR or other spectroscopic technique, SEM/EDS).
- Answer: New FTIR spectra were added to identify the curing mechanism beside the TGA-DTG thermograms.
Reviewer 2 Report
Reviewer’s report on MS #Coatings-759877
“Functionalization of Silica with Triazine Hydrazide to Improve Corrosion protection and Interfacial Adhesion Properties of Epoxy Coating and Steel Substrate”
This manuscript reports an approach for the modification of the epoxy coatings by silica with triazine hydrazide. The topic and key findings can be of interest, but the manuscript is neglectfully prepared and should be significantly improved before it can be published.
Below are listed my comments:
- The quality of the presentation should be improved. The manuscript has numerous typos and grammatical errors. It should be edited and revised.
- All images are of very low quality and neglectfully prepared. They are far beyond the standards of publishing, especially SEM.
Fig.2: All images are of low quality and stretched.
Fig. 5: All images are of low quality, stretched, different magnifications. It is impossible to compare the data and make any conclusions from them.
Fig.7: All images are of different sizes, no scale bar is provided.
Fig. 7a has been already published (DOI 10.1016/j.porgcoat.2020.105549). In the original article, it is stated as a sample after 400 h of corrosion tests, while in this manuscript this image shows the surface after 500 h of tests…
Schemes 1 and 2 report S, not Si, as i is barely visible.
- It is unclear for me what are Card-TH2 and CardT-TPA samples mentioned in the capture of Table 2.
- All the stages in schemes 1 and 2, or at least the final products, should be proved by the NMR spectroscopy results. Presented FTIR spectra are not fully convincing for me.
- To better evaluate the improvement in the corrosion protection authors are advised to perform EIS tests and especially evaluate the water uptake rates of the blank and modified coatings.
Author Response
- The quality of the presentation should be improved. The manuscript has numerous typos and grammatical errors. It should be edited and revised.
Answer: The article revised and the new corrections nd clarification marked with the red colour
- All images are of very low quality and neglectfully prepared. They are far beyond the standards of publishing, especially SEM.
Answer: The images were modified.
Fig.2: All images are of low quality and stretched.
Answer: The quality modified
Fig. 5: All images are of low quality, stretched, different magnifications. It is impossible to compare the data and make any conclusions from them.
Answer: The sizes of figures small and the images quality and scale bare clarified
Fig.7: All images are of different sizes, no scale bar is provided.
Answer: the scale bare modified.
Fig. 7a has been already published (DOI 10.1016/j.porgcoat.2020.105549). In the original article, it is stated as a sample after 400 h of corrosion tests, while in this manuscript this image shows the surface after 500 h of tests…
Answer: The test carried here in epoxy solvent based which increased the salt spray resistance than the published before.
Schemes 1 and 2 report S, not Si, as i is barely visible.
Answer: It was modified
- It is unclear for me what are Card-TH2 and CardT-TPA samples mentioned in the capture of Table 2.
Answer: No typing mistkes
- All the stages in schemes 1 and 2, or at least the final products, should be proved by the NMR spectroscopy results. Presented FTIR spectra are not fully convincing for me.
Answer: It is not easy to do NMR analysis for silica lower solubility and dispersion of the products in organic solvents
- To better evaluate the improvement in the corrosion protection authors are advised to perform EIS tests and especially evaluate the water uptake rates of the blank and modified coatings
Answer: The instrument not available at this time and I mentioned that salt spray can use to investigate the corrosion of steel in marine environment
Reviewer 3 Report
This paper reports an experimental study aimed to improve the corrosion resistance and interfacial adhesion of steel coatings based on epoxy. Specifically, silica gels were activated by Triazine Hydrazide by following a novel procedure of surface functionalization, which is properly described.
This technical paper contains interesting and original results, the accuracy of the measurements is high. The manuscript is well structured, abstract is coincise and enougly informative, the state-of the art is properly introduced, the results are well introduced, described and discussed.
I just have only few remarks, namely:
Aiming to help the readness and the understanding of some parts of the manuscript, I strongly suggest the authors for an extensive editing of English by a native speaker and to check the manuscript carefully before the submission for removing typos, etc. Following some examples:
Line 13: “is advanced approach” replaced by "is an advanced approach"
Line 32: “to remove the contaminates on the steel surface” replaced by “ to remove the contaminates from the steel surface"
Line 37: “New approach based on modification the chemical structure” replaced by “ New approach based on modification of the chemical structure”
When it is possible, please try to avoid long sentences.
Line 67-73: Please correct the font
Why STH in bracket? Please extend the terms at the first their appearance or add a list of symbols.
“Application of coatings of the steel substrate” as title. The term “Application” may be replaced by “deposition”
Line 139-“were elucidated” replaced by “were evinced”
Line 321: Cried out? I suppose “carried out”
At last, the conclusions should be structured as bullet points.
The paper may be published after a minor revision, in accordance with the comments and suggestions made to the authors.
Author Response
AI strongly suggest the authors for an extensive editing of English by a native speaker and to check the manuscript carefully before the submission for removing typos, etc. Following some examples:
Answer: ew corrections added and marked with red colour
Line 13: “is advanced approach” replaced by "is an advanced approach"
Answer: corrected
Line 32: “to remove the contaminates on the steel surface” replaced by “ to remove the contaminates from the steel surface"
Answer: changed
Line 37: “New approach based on modification the chemical structure” replaced by “ New approach based on modification of the chemical structure”
answer: new sentence added
When it is possible, please try to avoid long sentences.
Line 67-73: Please correct the font
answer: they modified
Why STH in bracket? Please extend the terms at the first their appearance or add a list of symbols.
“Application of coatings of the steel substrate” as title. The term “Application” may be replaced by “deposition”
The application changed to deposition
Line 139-“were elucidated” replaced by “were evinced”
answer: All changed
Line 321: Cried out? I suppose “carried out”
Answer: it is corrected
At last, the conclusions should be structured as bullet points.
Answer: The contineous conclusion recommended for short conclusions.
Round 2
Reviewer 2 Report
I thank the authors for their reply to manuscript Coatings 759877.
Unfortunately, most of my comments remained unconsidered and I did not find convincing rebuttal from the authors.
Please see the attached report.
I am sorry but I cannot recommend publication of this manuscript due to ethical concerns about some data and lack of supporting measurements for the corrosion section.

Author Response
Questions remained from Review #1 1. Indeed, the authors modified the text. However, the level of English is still very low, and the manuscript has a lot of typos and grammar errors. It should be revised again.
Answer: New corrections were marked with the red colour.
- Schemes 1 and 2 report S, not Si, as i is barely visible. Answer: It was modified My comment: I still see S, not Si in schemes 1 and 2. The grey rectangle for SG and AS is located wrong. What is its meaning at all? Please see the printscreen below, where I marked the problem.
Answer: Scheme 1 and 2 corrected to avoid the problems
Fig. 5: Your SEM images are still presented in different magnifications. How can a reader compare the morphology of your coating if you present x30000 for 3 wt% and x900 for 5 wt%. The same goes for different fillers. For me it is hard to evaluate the effect of different filler if one image shows 10 µm and other 150 µm of the surface area. Please unify this and make them all the same. I also have a comment regarding the image (b) for 3 wt%. In the original submission you presented it as x20000 magnification, while in the revised version it is x30000 (see below). This information is automatically added by the SEM equipment. For me it arises a major concern on the reliability of the data presented by the Authors in the examined manuscript.
Answer: The x is magnification factor and SEM used to determine the surface morphologies and cannot used to determine the particle sizes which determined by dynamic light schattering Figure 4. Regarding to image b The magnification x 20, 000 replaced with 30, 000 using different scale bare. I added The new figures at operating voltage 15 KV and I cannot add figures having same magnification factor.
- It is still unclear for me what are Card-TH2 and CardT-TPA, which have been added to the coatings examined in this Table. This caption is the only place in the manuscript where abbreviations Card-TH2 and CardT-TPA are mentioned. Authors do not describe it neither in the experimental, no results and discussion part. Please, provide explanation of these abbreviations.
Answer: The caption corrected
- I am sorry, but I did not understand the reply. I still see the ethical conflict here. The authors present the same image (which is not a good practice), while also state that here this image illustrates the surface after 500 h of tests. In the original paper it was claimed as the surface after 400 h of tests. For me remained unclear how the same image can represent the surface after 400 h and 500 h, whatever the media used. Again, for me it arises a major concern on the reliability of the data presented by the Authors in the examined manuscript… See the comparison below
Answer: I added new figures for the blank because it is the same blank published before.
My comment: I agree that salt spray tests are useful to evaluate the corrosion resistance. However, in the present form the mechanism proposed by the Authors (microcracks, retardation of steel dissolution) is speculative and not based on any experimental results. This should be improved and additional experiments (EIS) are needed.
Answer: The work stopped here extended for 21 days and we cannot do EIS measurements which replaced in several work by salt spray
You now have two Fig. 7 (see lines 254 and 336).
Answer: It is corrected
- You have used very toxic substances for your synthesis, which is not very environmentally friendly. It goes for the synthesis itself and possible remnant amounts in the formed materials. They can then be transferred to the environment. How did you clean the product? This aspect must be commented in the manuscript.
Answer: The materials prepared in reaction flak equipped with reflux condenser and the products washed with ethanol several times.
- Fig.1 is still of a very low quality. Please improve it.
Answer: The quality improved.
- Line 119, section 2.5. There is no description of the salt spray resistance here. Please correct the title.
Answer: It is corrected
- If you used sea water you must provide its composition, source, etc.
Answer: The source and specification inserted in the experimental
6. Line 130. Cations of metals should be marked as Men+ , not Me+n. In the present form you report the oxidation state not the charge of the cation.
Answer: They are corrected.
- New Fig. 5. Please indicate peaks and correct spelling on y-axis. I think you do not need the numbers on y-axis either
Answer: The curves closed to other the peaks assigned in Figure 1
Round 3
Reviewer 2 Report
Some punctuation and grammar problems still exist, I guess they would be fixed on the proof stage.